# A Comprehensive Analysis of Tn and STn Antigen Expression in Esophageal Adenocarcinoma

**DOI:** 10.3390/cancers16020240

**Published:** 2024-01-05

**Authors:** Baris Mercanoglu, Karl-Frederick Karstens, Anastasios D. Giannou, Jan Meiners, Jöran Lücke, Philipp Seeger, Vera Brackrock, Cenap Güngör, Jakob R. Izbicki, Maximilian Bockhorn, Thilo Hackert, Nathaniel Melling, Gerrit Wolters-Eisfeld

**Affiliations:** 1Department of General, Visceral and Thoracic Surgery, University Medical Center Hamburg-Eppendorf, 20246 Hamburg, Germanya.giannou@uke.de (A.D.G.); j.luecke@uke.de (J.L.); c.guengoer@uke.de (C.G.); izbicki@uke.de (J.R.I.); maximilian.bockhorn@uol.de (M.B.); t.hackert@uke.de (T.H.); n.melling@uke.de (N.M.); 2Section of Molecular Immunology und Gastroenterology, I. Department of Medicine, University Medical Center Hamburg-Eppendorf, 20246 Hamburg, Germany; 3Department of Pathology, University Medical Center Hamburg-Eppendorf, 20246 Hamburg, Germany; 4Department of General and Visceral Surgery, University Medical Center Oldenburg, 26133 Oldenburg, Germany

**Keywords:** esophageal adenocarcinoma, EAC, Tn antigen, STn antigen, glycosylation

## Abstract

**Simple Summary:**

Esophageal adenocarcinoma is a type of cancer that originates in cells lining the lower part of the esophagus, the tube that connects the throat to the stomach. This cancer specifically arises from glandular cells, which produce mucus to help lubricate the lining of the esophagus. In this work, we investigated whether and to what extent truncated O-glycans, called Tn and STn antigens, are expressed under pathophysiological conditions and whether these correlate with clinicopathological data and overall survival. This investigation is particularly significant in the context of targeted therapies, since Tn and STn antigens are tumor-specific target structures.

**Abstract:**

Differential glycosylation, marked by the presence of truncated O-glycans, is a distinctive feature of epithelial-derived cancers. However, there is a notable gap in research regarding the expression of Tn and STn antigens in esophageal adenocarcinoma (EAC). To address this, we employed commercially available antibodies, previously validated for Tn and STn antigens, to analyze two cohorts of EAC tissues. Initially, large-area tissue sections from formalin-fixed paraffin-embedded (FFPE) EAC and corresponding healthy tissues were subjected to immunohistochemistry (IHC) staining and scoring. Subsequently, we evaluated the RNA expression levels of crucial O-glycosylation related genes—*C1GALT1* and *C1GALT1C1*—using a quantitative real-time polymerase chain reaction (qRT-PCR). In a comprehensive analysis, a substantial cohort of EAC tissues (*n* = 311 for Tn antigen, *n* = 351 for STn antigen) was investigated and correlated with clinicopathological data. Our findings revealed that Tn and STn antigens are highly expressed (approximately 71% for both) in EAC, with this expression being tumor-specific. Notably, Tn antigen expression correlates significantly with the depth of tumor cell infiltration (*p* = 0.026). These antigens emerge as valuable markers and potential therapeutic targets for esophageal adenocarcinoma.

## 1. Introduction

Esophageal cancer poses a significant threat, exhibiting a grim prognosis even with treatment interventions. Esophageal adenocarcinoma (EAC), one of the two primary histological types of esophageal cancer alongside squamous cell carcinoma, is particularly formidable due to its aggressive nature and frequent late-stage detection, contributing to its relatively poor prognosis [1].

The genesis of EAC is commonly associated with Barrett’s esophagus, a condition where normal squamous epithelial cells in the lower esophagus transform into columnar cells resembling those in the stomach or intestines [2]. This transformation is often prompted by chronic gastroesophageal reflux disease (GERD), where stomach acid regurgitates into the esophagus. Various risk factors, including GERD, obesity, smoking, and male gender, are linked to the development of EAC [3]. Notably, the incidence of this cancer has been on the rise in Western countries, particularly in the United States and parts of Europe [4].

Diagnosis typically involves endoscopic procedures such as esophagogastroduodenoscopy (EGD) and biopsy to scrutinize the esophagus and confirm the presence of cancerous cells. Treatment modalities encompass surgery, chemotherapy, radiation therapy, or a combination thereof, contingent upon the cancer’s stage and extent [5].

Since the initial proposal connecting specific glycan structures to clinical cancer prognosis [6], there has been a significant upswing in interest in conducting studies aimed at identifying glycans and related glycoproteins present on the surfaces of cancer cells [7]. In the case of EAC, however, the availability of data on the expression of Tn and STn antigens and their associated clinical implications are limited.

Differential glycosylation can exert a significant influence on essential processes within cancer, with frequent observations of alterations in O-glycosylation [8]. In principle, proteins undergo diverse and abundant post-translational modification through O-glycosylation in the Golgi apparatus, a process highly dependent on the sequential activity of multiple glycosylation enzymes. The initiation of O-glycan synthesis is orchestrated by members of the GalNAc-transferase family, transferring N-Acetylgalactosamine (GalNAc) to serine or threonine residues on proteins (GalNAcα-O-Ser/Thr), leading to the formation of Tn antigen (CD175). Following initial processing, subsequent steps entail the branching and capping of the Tn antigen through the actions of various glycosyltransferases, like T-Synthase (*C1GALT1*) or Core 3 Synthase (*B3GNT6*). In healthy cells, O-glycosylation proceeds to yield mature, elongated, and branched O-glycans, often subject to modification with sialic acid. The specific patterns of glycans manifested in a cell are contingent upon the expressed glycosyltransferases, their substrate specificities, and cellular localization. In cancer cells, the progression of O-glycans into mature branched structures is frequently disrupted, resulting in the expression of early biosynthetic intermediates [9]. The sialyl-Tn antigen (Neu5Acα2-6GalNAcα-O-Ser/Thr), commonly known as STn, constitutes a truncated O-glycan characterized by a sialic acid α-2,6 linkage to GalNAc α-O-Serine/Threonine. The enzymatic participation of ST6GALNAC1 is essential for the formation of the STn antigen [10]. This glycan variant is known for its role in promoting cancer progression as it induces oncogenic properties in cancer cells. These features include increased proliferation, altered tissue architecture, impaired basement membrane adhesion, and the promotion of invasive growth [11].

Tn and STn neo- or over-expression has been reported in various epithelial- and non-epithelial-derived cancers, including colon, breast, skin, lung, pancreas [12], cervical, gastric, prostate, bladder, endometrial [13], and squamous esophageal cancers [14], and are associated with an unfavorable prognosis in cancer patients. Additionally, there is compelling evidence suggesting that molecular alterations in O-glycosylation may be intricately linked to the signaling pathways driving the malignant transformation of cells, thereby contributing to the malignant phenotype [15,16]. An elevated expression of the STn antigen in patients with gastric cancer has been consistently linked to a significantly unfavorable prognosis, as evidenced by independent studies [17,18]. The STn antigen, present in over 80% of human carcinomas, exhibits a close association with adverse outcomes in cancer cases. Notably, a prevalence of sialoglycoforms over neutral glycoforms has been reported in bladder tumors. Of particular interest, advanced bladder tumors with MUC16-STn glycoforms have been identified, showing a correlation with a worse prognosis [19]. The detection of the STn antigen on MUC16 serves as a valuable diagnostic tool, enabling the discrimination between endometriosis and ovarian cancer [20].

In various cancers, including gastric, colorectal, pancreatic, cervical, endometrial, and ovarian cancers [21,22,23,24,25], elevated levels of the STn antigen in serum have been documented. This elevation is consistently correlated with factors such as tumor size and metastasis [26,27]. In prostate cancer, the STn antigen is identified in up to half of all high-grade tumors [28], and the correlation of STn-MUC1 with survival outcomes and increased serum PSA levels has been established [29]. Furthermore, a recent study has expanded the understanding by associating heightened serum STn antigen levels with histological grade and lymph node metastasis in endometrial cancer [21].

Given the prevalence of glycans on the outer layer of cancer cells and the distinct expression of certain glycan structures such as the Tn and STn antigens across various tumor types, these molecules hold potential as targets for disease diagnosis, prognosis assessment, and drug delivery.

## 2. Materials and Methods

### 2.1. Patient Selection and IHC Establishment

A total of 36 patient-derived esophageal cancer biopsies and corresponding healthy tissues were selected from our cryo-biobank of native tissues. Only tissues with histologically confirmed advanced adenocarcinomas were selected for RNA isolation (see real-time PCR) and FFPE treatment. For IHC scoring of Tn and STn antigens, tumors with complete absence of staining or weak staining intensity with <10% positivity, no surface staining, or pure cytosolic staining were scored as “negative” with a score of 0. All tumor samples with ≥10% positivity, and with membranous accentuation and homogenous staining of the tumor tissues, were accordingly either scored as “weak”, “moderate”, or “strong”, depending on the staining intensity (scores of 1–6).

The procurement of tissue specimens and their subsequent analysis for research endeavors received approval from the Medical Ethical Committee in Hamburg, Germany (PV3548). Prior to inclusion in the study, informed consent was obtained from all participating patients. The entirety of the work adheres to the principles outlined in the Declaration of Helsinki.

### 2.2. Esophageal Cancer Tissue Microarray

The esophageal cancer TMA utilized for this study consists of 1011 formalin-fixed paraffin-embedded tissue samples including 501 esophageal adenocarcinomas and 510 esophageal squamous cell carcinomas (not included here), and was extended based on an earlier TMA containing 292 cancers, as previously described by us [30,31]. All patients had undergone surgery between 1992 and 2011 at the Department of General, Visceral and Thoracic Surgery at the University Hospital Hamburg-Eppendorf. All work has been carried out in compliance with the Declaration of Helsinki. The general usage of archived diagnostic leftover tissues for manufacturing of tissue microarrays (TMAs) and their analysis for research purposes as well as patient data analysis has been approved by the Ethics Committee Hamburg (WF-049/09 and PV3652). The TMA manufacturing process was described earlier in detail [32]. In short, one 0.6 mm core was taken from a representative tissue block from each patient. Tissue samples were distributed on 4 TMA blocks, containing 369, 254, 292, and 96 cancer cores, respectively. In addition, all blocks comprised tissue controls of normal esophageal epithelium. Tumor grade and stage were defined according to the International Union Against Cancer (UICC) and the WHO [33,34]. Clinical data of patients were retrospectively evaluated. An overview of all demographic and clinical data is given in Table 1.

### 2.3. Immunohistochemistry

Freshly cut TMA sections were immuno-stained on one day and in one experiment. Slides were deparaffinized and exposed to heat-induced antigen retrieval for 5 min in an autoclave at 121 °C in pH 2.0, in target retrieval solution (Biogenex, San Ramon, CA, USA). Primary antibodies specific for Tn antigen/CD175 (mouse monoconal IgG, dilution 1:150, Thermo Fisher Scientific, Waltham, MA, USA, cat. number MA1-80055) and STn antigen/CD175s (mouse monoclonal IgG, dilution 1:150, Thermo Fisher Scientific, Waltham, MA, USA, cat. number MA1-90577) was applied at 37 °C and pH 7.8 for 60 min. Bound antibody was then visualized using the EnVision Kit (Dako, Glostrup, Denmark) according to the manufacturer’s directions. CD175 and CD175s staining was found in the cytoplasm and the apical membrane of positive cells. Tumors with complete absence of staining were scored as “negative”. Tumor samples with ≥10% positivity, and with membranous accentuation, displaying a homogenous staining of the complete core were accordingly scored as “positive”.

### 2.4. Real-Time qRT-PCR

Total RNA was extracted from EAC and corresponding healthy tissue using TRIzol reagent (Invitrogen, Waltham, MA, USA). The high-capacity cDNA synthesis Kit (Applied Biosystems, Waltham, MA, USA) was used for cDNA synthesis. Real-time PCR was performed using the Kapa Probe Fast qPCR Master Mix (Kapa Biosystems, Wilmington, MA, USA) on the StepOnePlus system (Applied Biosystems, Waltham, MA, USA). Probes used were *HPRT1* (Hs02800695_m1), *C1GALT1* (Hs00750511_s1), and *C1GALT1C1* (Hs04999676_s1), all from Thermo Fisher Scientific. Relative RNA expression was normalized to *HPRT1* and calculated using the 2^−ΔΔCt^ method.

### 2.5. Statistical Analysis

Survival intervals were calculated from the time of surgery to the occurrence of disease-related death. Contingency tables and the chi²-test were employed to explore potential associations between molecular parameters and tumor phenotype. Kaplan–Meier curves were generated to estimate survival, and a log-rank test was conducted to compare survival variables in the univariate analysis. Data analysis was carried out using SPSS software (Version 25), with all tests being two-sided. The significance level was set at 0.05. Violin plots and bar graphs were generated using GraphPad Prism 9.5.1. The significance of comparative real-time qRT-PCR was determined using a two-tailed *t*-test.

## 3. Results

### 3.1. Esophageal Adenocarcinoma Express High Levels of Tn and STn Antigen

Initially, we performed immunohistochemistry analysis to comprehensively characterize the presence of Tn and STn antigens in a cohort comprising 36 patient-derived EAC samples and their respective healthy tissue counterparts. Our results revealed high expression levels of Tn and STn in tumoral tissues compared to their healthy esophageal counterparts. Both Tn and STn antigens exhibited prominent surface expression in a substantial fraction of EAC specimens (Figure 1a,b). We categorized immunohistochemistry-stained sections for Tn and STn antigens as either positive (with weak, medium, or high staining intensity, ≥10% positivity, and membranous accentuation) or negative (with weak staining intensity, <10% positivity, no surface staining, or pure cytosolic staining). All examined EAC tissues displayed varying degrees of expression for both Tn and STn antigens. Notably, Tn antigen reactivity in healthy tissues was observed in only one case (Figure 1c), while the STn antigen was present in seven healthy tissues, but was limited to suprabasal esophageal cells and the keratin layer (Figure 1d).

To elucidate the mechanism behind truncated O-glycan formation in EAC, we compared the gene expression of *T-Synthase* (*C1GALT1*) and its chaperone *COSMC* (*C1GALT1C1*) in healthy and EAC tissues. A modest, yet statistically significant upregulation in T-Synthase expression was observed in the tumor compared to controls. Conversely, the expression of *C1GALT1C1* was notably reduced in the tumor compared to normal esophagus tissues (Figure 1e,f).

### 3.2. Demographic and Histopathological Parameters

The EAC patient cohort comprised 76 (15.2%) females and 425 (84.8%) males. Predominantly, patients were diagnosed in tumor stage pT3 (*n* = 278, 55.5%), with a majority exhibiting moderate differentiation (*n* = 318, 63.3%), and a significant proportion displaying poor differentiation (*n* = 169, 33.7%). Notably, lymph node metastasis was detected in the majority of cases (*n* = 366, 73.1%), while metastasis to other parts of the body was notably absent in a significant majority of cases (*n* = 421, 84.0%), as detailed in Table 1.

To delineate the roles of Tn and STn antigens, we analyzed cancer tissue obtained from 501 esophageal adenocarcinoma patients. A total of 311 adenocarcinomas were deemed interpretable for our TMA analysis of the Tn antigen, and 353 cases were assessable for the STn antigen. Non-informative cases were attributed to either a complete absence of tissue or the lack of unequivocal cancer tissue in the TMA section.

### 3.3. Tn Antigen Immunostaining in Esophageal Cancers

Tn antigen immunostaining was absent in benign esophageal tissue and was predominantly localized to the cell membrane in esophageal adenocarcinoma. Notably, Tn antigen expression was detectable in 71% of all cases. Representative images of both weak and strong Tn antigen immunostaining in cores of the EAC TMA are depicted in Figure 2.

### 3.4. The Association between Tn Antigens and Histopathological Tumor Phenotype

Within the EAC cohort, the depth of infiltration (*p* = 0.026) demonstrated a significant association with Tn antigen expression, while UICC staging (*p* = 0.073) exhibited a tendency toward a correlation with Tn antigen expression. Conversely, lymphatic (*p* = 0.288) and distant metastasis (*p* = 0.258), differentiation (*p* = 0.246), and resection margin status (*p* = 0.726) showed no significant correlation with Tn antigen expression, as outlined in Table 2.

### 3.5. STn Antigen Immunostaining in Esophageal Cancers

STn antigen immunostaining was typically absent in benign esophageal tissue and was predominantly localized to the cell membrane in esophageal adenocarcinoma. Notably, STn antigen expression was detectable in 70.5% of all cases. Representative images illustrating both weak and strong STn antigen immunostaining in the TMA cores of EAC are presented in Figure 2.

### 3.6. The Association between STn Antigens and Histopathological Tumor Phenotype

In EAC, no significant correlation was identified with infiltration depth (*p* = 0.122), lymphatic (*p* = 0.887) or distant dissemination (*p* = 0.506), tumor stage (*p* = 0.581), resection margin (*p* = 0.761), or differentiation (*p* = 0.848), as detailed in Table 3.

### 3.7. Tn and STn Antigen and Overall Survival

Follow-up data were accessible for 311 and 353 EAC patients with informative Tn and STn antigen data, respectively. Notably, Tn and STn antigen immunostaining exhibited no significant association with the overall survival of EAC patients (*p* = 0.562 and *p* = 0.171). The correlation between Tn and STn antigen immunostaining and the clinical outcome of the patients is depicted in Figure 3a,b.

## 4. Discussion

In this study, a large cohort of esophageal adenocarcinoma was stained against Tn and STn antigens using immunohistochemistry. Both tumor-associated O-glycans were expressed at 71% (Tn antigen) and 70.5% (STn antigen), across all tumor stages and grades. The survival analysis conducted in this study did not identify a statistically significant correlation between the expression of Tn and STn antigens and the overall survival of the patients. This finding implies that these antigens may have limitations when utilized as independent prognostic indicators and may be more appropriately regarded as general tumor markers in the context of EAC. Our retrospective study aims to serve as an initial exploration, paving the way for further analyses of Tn and STn antigens within the framework of clinicopathological correlation. Subsequent international research efforts may contribute to mitigating potential general biases and enriching our understanding of these antigens in EAC. The use of commercially available antibodies in this study guarantees the reproducibility of analyses across diverse cohorts, thereby advancing our comprehension of Tn and STn antigen expression in esophageal adenocarcinoma (EAC) and other tumor pathologies. It is essential to acknowledge, however, that monoclonal antibodies (mAbs) specific to Tn and STn antigens capture only a fraction of the tumors’ complexity. Consequently, future investigations should contemplate employing a spectrum of antibodies targeting various glyco-epitopes, along with the potential inclusion of well-characterized plant lectins or recombinant glycan-binding receptors. Such a multifaceted approach holds promise for a more comprehensive exploration of the intricacies within tumor biology.

To decipher the composition of the O-glycome, the COSMC gene was selectively deactivated in numerous human cancer cell lines through gene editing techniques, inducing the expression of the Tn antigen. Subsequently, protein and glycopeptide isolation was achieved using the Tn antigen-affine plant lectin Vicia villosa agglutinin (VVL), followed by identification through mass spectrometric proteome analysis [35]. Notably, the discernment emerged that there exists no discernible overlap in the identified O-glycoproteins, even when comparing cell lines within the same tumor entity. This revelation underscores the considerable heterogeneity inherent in the O-glycoproteome, introducing complexity into cell biological processes that defy straightforward predictions. Importantly, there remains a notable absence of an O-glycoproteome analysis for EAC to date.

The analysis of mRNA expression for pivotal genes involved in the initial step of O-glycosylation reveals a substantial downregulation of *C1GALT1C1* (*COSMC*) in the carcinomas. In contrast, there is a modest yet significant upregulation observed in *C1GALT1* (*T-Synthase*). The enzymatic role played by T-Synthase is crucial for the development of truncated O-glycans within tumors. T-Synthase requires its chaperone, COSMC, to convert the Tn antigen into the core 1 structure, which reflects the initial step in complex O-glycosylation. COSMC facilitates proper oligomerization and localization in the endoplasmic reticulum (ER). If COSMC undergoes mutation or depletion, it leads to the degradation of T-Synthase and the expression of Tn antigens [36].

Cummings et al. devised an assay for assessing T-Synthase activity in vitro, enabling the quantification of core 1 O-glycan synthesis in cell lines [37]. In the evaluation of T-Synthase activity in formalin-fixed, paraffin-embedded (FFPE) tissues, a challenge arises from the fact that the immunohistochemical detection of T-Synthase and COSMC offers only partial insights into protein detection, lacking information about their actual functionality. The same limitation applies to mRNA expression data, which currently does not facilitate the genotype–glycotype correlation.

It has been documented that the elevated expression of *C1GALT1* contributes to the increased O-glycosylation of Mucin-1 (MUC1) in patients with esophageal squamous cell carcinoma [38] and enhances the invasive and metastatic phenotype by modifying O-glycans on integrin β1 in hepatocellular carcinoma [39]. Additionally, a complete gene knockout of *C1galt1c1* in a pancreatic acinar mouse model resulted in the elevated expression of *C1galt1* [40]. In Tn4 cells (Epstein–Barr virus (EBV)-transformed B lymphocytes from a male individual whose leukocytes express Tn antigens), characterized by silenced *COSMC*, methylome analysis unequivocally confirmed the hypermethylation of the *COSMC* core promoter, with no such observation noted for *T-Synthase* [41]. The transcriptional regulation of *COSMC* and *T-Synthase*, along with their widespread and coordinated expression patterns, requires further clarification in the context of EAC [42]. Additional investigation into glycan-related pathways and cell-to-cell interactions within the tumor microenvironment is essential to elucidate the involvement of T-Synthase and its associated molecules in the progression of tumors.

In a recent study, the presence of truncated O-glycans was assessed across various epithelial and non-epithelial cancers using antibodies targeting the Tn antigen (5F4), the STn antigen (3F1), and the T antigen (3C9). In the context of gastrointestinal tumors, the expression of Tn and STn antigens was detected in pancreatic adenocarcinoma (PDAC) (53%/56%) and colorectal adenocarcinoma (CRC) (51%/80%). Interestingly, the authors noted that Tn and STn antigens were predominantly expressed in tumor tissue and sporadically in healthy tissues [12]. Our data similarly revealed that the Tn antigen antibody yielded positive staining in only one case (*n* = 30), while the STn antigen antibody displayed positive staining in healthy tissue in 5 out of 30 cases. This suggests the transferability of the antibody specificities and an overarching robustness of the presented results.

In the case of PDAC, it was demonstrated that the aberrant expression of truncated O-glycans in PDAC enhances tumor aggressiveness by inducing epithelial-mesenchymal transition (EMT) and stemness properties [43,44].

For CRC, minimal intracellular Tn staining was observed in normal mucosa, while a significantly higher expression was noted in both peritumoral mucosa and adenocarcinoma. This staining pattern was reflected to a lesser extent by STn expression in these tissue types [45].

The glycocode not only governs immune cell lectin recognition, steering the immune response toward cancer-tolerant phenotypes [46], it also exerts influence within the tumor microenvironment. This environment promotes glycophenotypes that facilitate disease progression and dissemination, although the precise molecular and clinical drivers of glycosylation changes remain elusive in many instances [47]. The rapid translation of communication between the tumor microenvironment and tumor cells occurs through dynamic alterations in cell surface glycosylation, contributing to stress adaptation or active evasion [48]. Another pivotal consideration is the intricate interplay between different O-glycans within both the tumor and immune system cells. This dynamic interaction is critical in the context of glycoprotein modifications on the cell surface, influencing recognition by glycan-binding receptors such as C-type lectins, selectins, siglecs, and galectins. As glycans have evolved into essential components within homeostatic circuits, functioning as precise modulators of immunological responses, they emerge as potential molecular targets for intentionally modulating immune tolerance and activation across a spectrum of pathological scenarios. The recent review article emphasizes the significant role of Tn and STn antigen expression in gastrointestinal tumors in contributing to immunosuppression [49]. The focal point remains on innate immune system cells, specifically those expressing glycan-binding receptors like CD301 (MGL) [50], Siglec-4, -5, -14, and -15 [51], and potentially other receptors selectively interacting with tumor O-glycans.

To sum up, glycans serve as integral mediators of both external and internal signals, resulting in phenotypic changes on the cell surface and actively contributing to malignancy, or serving as surrogate markers for cellular adaptation. Consequently, glycans emerge as highly promising targets for theragnostic applications.

The Tn and STn antigens, recognized as truncated O-glycans, are well-defined cancer-associated glycans expressed across a spectrum of epithelial cancers, albeit to varying degrees. Consequently, therapeutic approaches targeting truncated O-glycans have surfaced, suggesting potential relevance in the future treatment of esophageal adenocarcinoma. A range of promising active compounds falls within different categories, encompassing antibodies [52], antibody-drug conjugates [53], CAR T cells [54,55,56], and sialidases [57].

## 5. Conclusions

Our study revealed a pronounced expression of Tn and STn antigens in esophageal adenocarcinoma compared to corresponding healthy tissues. Immunohistochemistry analysis demonstrated prominent surface expressions of both antigens in a substantial fraction of EAC specimens, distinguishing them as specific markers for tumor tissues. Notably, a limited or absent expression of these antigens was observed in healthy esophageal tissues.

To decipher the mechanisms underlying truncated O-glycan formation in EAC, we assessed the expression of *T-Synthase* (*C1GALT1*) and its chaperone *COSMC* (*C1GALT1C1*). Tumor tissues exhibited a statistically significant upregulation in *T-Synthase* expression, while *C1GALT1C1* expression was notably reduced, compared to normal esophagus tissues, probably causing T-Synthase to malfunction.

Tn antigen expression demonstrated a significant association with the depth of infiltration, suggesting its potential role in tumor invasiveness. However, no significant correlation was observed with lymphatic or distant metastasis, tumor differentiation, UICC staging, or resection margin status. Similarly, STn antigen expression did not show a significant correlation with various histopathological parameters.

Analysis of overall survival in EAC patients with Tn and STn antigen immunostaining revealed no significant association, indicating that the expression of these antigens alone may not serve as independent prognostic indicators.

In summary, our comprehensive investigation establishes Tn and STn antigens as distinctive markers of esophageal adenocarcinoma. While Tn antigen expression correlates with the depth of infiltration, their association with overall survival suggests a complex interplay with other factors influencing the clinical outcome of EAC patients. These findings contribute valuable insights into the molecular landscape of esophageal adenocarcinoma, potentially informing future diagnostic and therapeutic strategies.

## Figures and Tables

**Figure 1 cancers-16-00240-f001:**
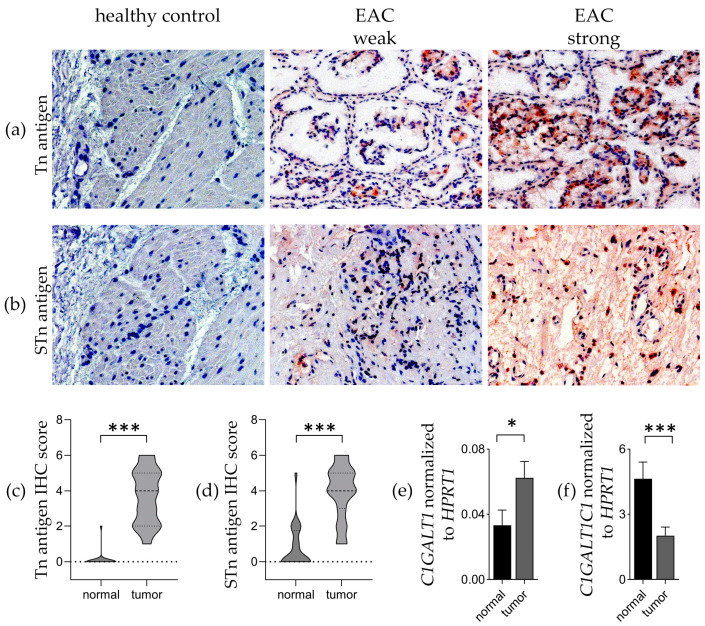
Truncated O-glycans Tn and STn antigens are highly expressed in EAC. (**a**,**b**) Representative microscopic images at ×100 magnification featuring immunohistochemical staining with monoclonal antibodies targeting Tn or STn antigens on sections of EAC (weak and strong) or healthy esophageal tissue. (**c**,**d**) Sores of Tn and STn antigen IHC staining reflecting antibody binding in normal and EAC tissue. (**e**) Analysis and quantification of *T-Synthase* (*C1GALT1*) mRNA and (**f**) *COSMC* (*C1GALT1C1*) expression levels in healthy and EAC tissues. * equals *p* ≤ 0.05 and *** equals *p* ≤ 0.001.

**Figure 2 cancers-16-00240-f002:**
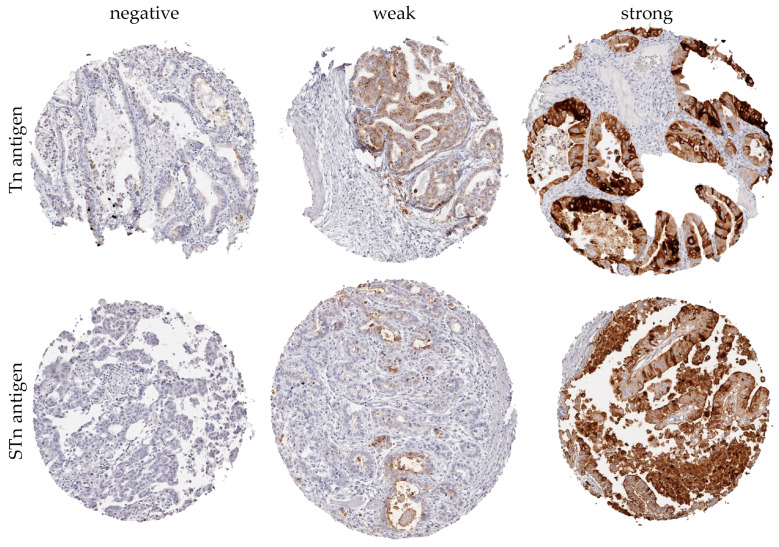
Microscopic images representing EAC TMA cores at ×100 magnification were obtained, featuring immunohistochemical (IHC) staining with monoclonal antibodies targeting Tn or STn antigens. The images capture varying degrees of Tn and STn antigen expression in EACs, ranging from negative to weak and strong.

**Figure 3 cancers-16-00240-f003:**
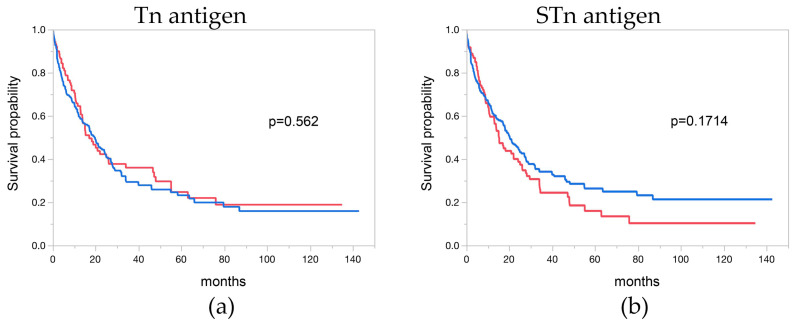
Overall survival of patients with esophageal adenocarcinoma based on Tn and STn antigen staining. (**a**) The Kaplan–Meier plot depicts EAC Tn antigen staining, and (**b**) STn antigen staining. The overall survival p-values indicate no significant difference between the analyzed groups, as determined by the LogRank test for Tn (*p* = 0.1714) and STn antigen (*p* = 0.562). The red line represents negative cases, while the blue line represents positive cases.

**Table 1 cancers-16-00240-t001:** Histopathological and clinical features of the arrayed EAC specimens.

Parameter		EACs (*n* = 501)
Sex	male	425
	female	76
Tumor stage	pT1	108
	pT2	87
	pT3	278
	pT4	28
Lymph node metastasis	pN0	135
	pN1	172
	pN2	95
	pN3	99
UICC	I	110
	II	87
	III	229
	IV	75
Distant Metastasis	M0	421
	M1	80
Grading	G1	14
	G2	318
	G3	169

**Table 2 cancers-16-00240-t002:** Association of Tn antigen (CD175) IHC results and clinicopathological features of EACs.

	Tn Antigen (CD175)			
Parameter	*n* Evaluable	Negative (%)	Positive (%)	*p* Value
All cancers	311	28.9	71.1	
Tumor stage				
pT1	51	31.4	68.6	0.0266
pT2	58	39.7	60.3	
pT3	182	26.4	73.6	
pT4	17	5.9	94.1	
Lymph node metastasis				
pN0	76	32.9	67.1	0.2877
pN1	107	28.0	72.0	
pN2	56	33.9	66.1	
pN3	60	20.00	80.00	
UICC				
I	55	30.9	69.1	0.0732
II	51	41.2	58.8	
III	157	26.1	73.9	
IV	40	17.50	82.50	
Distant metastasis				
M0	259	29.7	70.3	0.2579
M1	46	21.7	78.3	
Resection margin				
R0	219	29.7	70.3	0.7256
R1	72	25.00	75.00	
R2	8	25.00	75.00	
Grading				
G1	10	20.00	80.00	0.2463
G2	108	34.3	65.7	
G3	183	25.7	74.3	

**Table 3 cancers-16-00240-t003:** Association of STn antigen (CD175s) IHC results and clinicopathological features of EACs.

	STn Antigen (CD175s)			
Parameter	*n* Evaluable	Negative (%)	Positive (%)	*p* Value
All cancers	353	29.5	70.5	
Tumor stage				
pT1	59	25.4	74.6	0.1218
pT2	63	39.7	60.3	
pT3	208	28.8	71.2	
pT4	20	15.0	85.0	
Lymph node metastasis				
pN0	86	26.7	73.3	0.8868
pN1	117	30.8	69.2	
pN2	62	32.3	67.7	
pN3	76	28.9	71.1	
UICC				
I	62	25.8	74.2	0.581
II	56	35.7	64.3	
III	178	30.3	69.7	
IV	48	25.0	75.0	
Distant metastasis				
M0	293	30.4	69.9	0.5062
M1	54	25.9	74.1	
Resection margin				
R0	248	30.2	69.8	0.7611
R1	85	27.1	72.9	
R2	8	37.5	62.5	
Grading				
G1	12	25.0	75.0	0.8482
G2	120	27.5	72.5	
G3	210	30.0	70.0	

## Data Availability

The data that support the findings of this study are available from the corresponding author upon reasonable request.

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
