# Peer review of "A Comprehensive Analysis of Tn and STn Antigen Expression in Esophageal Adenocarcinoma"

_cancers, 2024, doi:10.3390/cancers16020240_

Round 1
Reviewer 1 Report
Comments and Suggestions for Authors
The study assessed Tn and sTn expression in esophageal adenocarcinoma mainly using the immunohistochemistry technique. It was found that both were highly expressed in the tumor tissue compared to the normal one. The staining pictures are of great quality. The tissue microarray they constructed is incredible, containing 501 EAC and 510 ESCC specimens. This is very useful in esophageal cancer studies. I wish to have some. Here are a couple of minor issues:
1. It is said in the section of Materials & Methods that RT_PCR was performed, but I do not see their results in the paper. Where are they?
2. On page 3, the subtitle "Esophageal cancer TMA" should be changed to "Esophageal cancer tissue microarray".
Author Response
On behalf of all contributing authors, I extend our gratitude for the positive reception of the manuscript and for your valuable comments. In the following sections, we address each of the reviewer's comments point by point.
The study assessed Tn and sTn expression in esophageal adenocarcinoma mainly using the immunohistochemistry technique. It was found that both were highly expressed in the tumor tissue compared to the normal one. The staining pictures are of great quality. The tissue microarray they constructed is incredible, containing 501 EAC and 510 ESCC specimens. This is very useful in esophageal cancer studies. I wish to have some. Here are a couple of minor issues:
- It is said in the section of Materials & Methods that RT_PCR was performed, but I do not see their results in the paper. Where are they?
Response: The findings from the real-time qRT-PCR expression analyses of C1GALT1 and C1GALT1C1 are depicted in Figure 1e + f. Reference to these results can be found in the revised manuscript on page 4, line 171: " To elucidate the mechanism behind truncated O-glycan formation in EAC, we com-pared the gene expression of T-Synthase (C1GALT1) and its chaperone COSMC (C1GALT1C1) in healthy and EAC tissues. A modest, yet statistically significant upregulation in T-Synthase expression was observed in the tumor compared to controls. Conversely, the expression of C1GALT1C1 was notably reduced in the tumor compared to normal esophagus tissues (Figure 1e-f)".
- On page 3, the subtitle "Esophageal cancer TMA" should be changed to "Esophageal cancer tissue microarray".
Response: In accordance with the reviewer's recommendation, we have expanded the abbreviation in the sub-heading.
Reviewer 2 Report
Comments and Suggestions for Authors
The aim of this work is to investigate the expression of Tn and STn antigens in esophageal cancer and their correlation with the clinical outcome. 501 patients with esophageal adenocarcinoma were subjected to immunohistochemical analysis of Tn and STn antigens for this study. To determine the function of Tn and STn antigens in esophageal adenocarcinoma, the expression, clinical characteristics, and survival outcomes of these antigens in these patients were analyzed. This study has the following primary issues, among others:
1) As a retrospective study, it is possible that there are biases in the information and patient selection processes.
2) Only using the immunohistochemical method to detect the expression of Tn and STn antigens may not fully reflect the real function and clinical significance of these antigens. The authors may need to verify the expression of Tn and STn antigens in other validation datasets.
3) The survival analysis of this study showed that there was no significant correlation between the expression of Tn and STn antigens and the total survival time of patients, which may indicate that these antigens may be limited as independent prognostic indicators.
4) This paper mentioned the key role of T-Synthase in the production of Tn and STn antigens. However, there is a lack of experimental verification of the possible mechanism between the activity of C1GALT1 and the occurrence and development of esophageal adenocarcinoma. Further analysis of glycans-related pathways and cell-to-cell interaction in tumor microenvironment is needed to clarify the role of C1GALT1 and its related molecules in tumor progression.
5) The correlation P value of resection margin at line 222 should be 0.761. Since the results in Table 3 and figure 3 are not statistically significant, it is suggested that they be included in the appendix, while the results of other data analyses recommended above are implemented in the main text.
Comments on the Quality of English LanguageThe quality of English language in this paper is commendable.
Author Response
Reviewer 2
On behalf of all contributing authors, I extend our gratitude for the favorable reception of the manuscript and for your invaluable and insightful comments. Your constructive feedback has significantly enhanced the content, providing readers with a clearer understanding to classify the results. In the subsequent sections, we address each reviewer's comment, addressing them point by point.
The aim of this work is to investigate the expression of Tn and STn antigens in esophageal cancer and their correlation with the clinical outcome. 501 patients with esophageal adenocarcinoma were subjected to immunohistochemical analysis of Tn and STn antigens for this study. To determine the function of Tn and STn antigens in esophageal adenocarcinoma, the expression, clinical characteristics, and survival outcomes of these antigens in these patients were analyzed. This study has the following primary issues, among others:
1) As a retrospective study, it is possible that there are biases in the information and patient selection processes.
Response: We have added the following statement to the discussion: “Our retrospective study aims to serve as an initial exploration, paving the way for further analyses of Tn and STn antigens within the framework of clinicopathological correlation. Subsequent international research efforts may contribute to mitigating potential general biases and enriching our understanding of these antigens in EAC.”
2) Only using the immunohistochemical method to detect the expression of Tn and STn antigens may not fully reflect the real function and clinical significance of these antigens. The authors may need to verify the expression of Tn and STn antigens in other validation datasets.
Response: The antibodies employed in this study have undergone comprehensive testing and prior publication, as exemplified in the work of Wolters-Eisfeld et al. (Exp Mol Med 2018). In that publication, we elucidated the specificities through immunohistochemical (IHC) analyses conducted on pancreatic tissue obtained from both wild-type and conditionally transgenic Cosmc-KO mice.
Formulating a suggested combination of detection reagents for Tn and STn antigens stands as a valuable initiative for establishing a robust foundation in the biomedical sector. This strategic approach would not only foster comparability but also ensure reproducibility of results, thereby enhancing the reliability of findings in our realm of research.
We have included the following passage in the discussion: “The use of commercially available antibodies in this study guarantees the reproducibility of analyses across diverse cohorts, thereby advancing our comprehension of Tn and STn antigen expression in esophageal adenocarcinoma (EAC) and other tumor pathologies. It is essential to acknowledge, however, that monoclonal antibodies (mAbs) specific to Tn and STn antigens capture only a fraction of the tumor's complexity. Consequently, future investigations should contemplate employing a spectrum of antibodies targeting various glyco-epitopes, along with the potential inclusion of well-characterized plant lectins or recombinant glycan-binding receptors. Such a multifaceted approach holds promise for a more comprehensive exploration of the intricacies within tumor biology.”
3) The survival analysis of this study showed that there was no significant correlation between the expression of Tn and STn antigens and the total survival time of patients, which may indicate that these antigens may be limited as independent prognostic indicators.
Response: The reviewer raises a crucial point in this context. We have added the following sentence to the discussion: “The survival analysis conducted in this study did not identify a statistically significant correlation between the expression of Tn and STn antigens and the overall survival of the patients. This finding implies that these antigens may have limitations when utilized as independent prognostic indicators and may be more appropriately regarded as general tumor markers in the context of EAC.”
4) This paper mentioned the key role of T-Synthase in the production of Tn and STn antigens. However, there is a lack of experimental verification of the possible mechanism between the activity of C1GALT1 and the occurrence and development of esophageal adenocarcinoma. Further analysis of glycans-related pathways and cell-to-cell interaction in tumor microenvironment is needed to clarify the role of C1GALT1 and its related molecules in tumor progression.
Response: We concur with the reviewer regarding this aspect. To address this concern, we have incorporated a new sentence in the discussion, which states "Additional investigation into glycans-related pathways and cell-to-cell interactions within the tumor microenvironment is essential to elucidate the involvement of T-Syntase and its associated molecules in the progression of tumors".
5) The correlation P value of resection margin at line 222 should be 0.761. Since the results in Table 3 and figure 3 are not statistically significant, it is suggested that they be included in the appendix, while the results of other data analyses recommended above are implemented in the main text.
Response: Thank you very much for pointing out the incorrect P value. We have corrected the relevant passage in the manuscript.
Given that the data provided in Table 3 and Figure 3 represents an initial description, we believe it is reasonable to display them prominently, even if none of the results are statistically significant.
Reviewer 3 Report
Comments and Suggestions for Authors
The study offers a thorough examination of Tn and STn antigen expression in esophageal adenocarcinoma (EAC), providing valuable insights into their potential as diagnostic markers. While the manuscript is well-constructed and the findings are compelling, it is essential to note that the observed expression of Tn and STn antigens has been previously reported in other cancers. While the novelty of these findings may be limited, their importance in the context of EAC cannot be understated. Here are a few minor suggestions to further refine the manuscript:
- In the introduction, the author should intricately explore into the clinical implications of Tn and STn antigens, particularly emphasizing their association with a poor prognosis. Provide a succinct overview of existing literature supporting this association, laying the foundation for the significance of these antigens in EAC.
- In the discussion section, the author should address the apparent discrepancy in the association between Tn and STn antigens and prognosis in EAC compared to other cancers. Explore and present the potential explanations or hypotheses behind this observed difference, shedding light on the intricate molecular and contextual factors specific to esophageal adenocarcinoma.
These revisions will enhance the manuscript by clarifying the clinical relevance of Tn and STn antigens in EAC and providing a nuanced understanding of why their prognostic implications might differ in this particular cancer type compared to others.
Author Response
Reviewer 3
On behalf of all contributing authors, I would like to thank you for the positive reception of the manuscript and for your valuable suggestions to improve our manuscript. Below we address the reviewer's comments.
The study offers a thorough examination of Tn and STn antigen expression in esophageal adenocarcinoma (EAC), providing valuable insights into their potential as diagnostic markers. While the manuscript is well-constructed and the findings are compelling, it is essential to note that the observed expression of Tn and STn antigens has been previously reported in other cancers. While the novelty of these findings may be limited, their importance in the context of EAC cannot be understated. Here are a few minor suggestions to further refine the manuscript:
In the introduction, the author should intricately explore into the clinical implications of Tn and STn antigens, particularly emphasizing their association with a poor prognosis. Provide a succinct overview of existing literature supporting this association, laying the foundation for the significance of these antigens in EAC.
Response: As suggested by the reviewer, we have added the following passage in the introduction: “Elevated expression of STn antigen in patients with gastric cancer has been consist-ently linked to a significantly unfavorable prognosis, as evidenced by independent studies [17,18]. STn antigen, present in over 80% of human carcinomas, exhibits a close association with adverse outcomes in cancer cases. Notably, a prevalence of si-aloglycoforms over neutral glycoforms has been reported in bladder tumors. Of par-ticular interest, advanced bladder tumors with MUC16-STn glycoforms have been identified, showing a correlation with worse prognosis [19]. The detection of STn an-tigen on MUC16 serves as a valuable diagnostic tool, enabling the discrimination be-tween endometriosis and ovarian cancer [20].
In various cancers, including gastric, colorectal, pancreatic, cervical, endometrial, and ovarian cancers [21-25], elevated levels of STn antigen in serum have been docu-mented. This elevation is consistently correlated with factors such as tumor size and metastasis [26,27]. In prostate cancer, STn antigen is identified in up to half of all high-grade tumors [28], and the correlation of STn-MUC1 with survival outcomes and increased serum PSA levels has been established [29]. Furthermore, a recent study has expanded the understanding by associating heightened serum STn antigen levels with histological grade and lymph node metastasis in endometrial cancer [21].”
In the discussion section, the author should address the apparent discrepancy in the association between Tn and STn antigens and prognosis in EAC compared to other cancers. Explore and present the potential explanations or hypotheses behind this observed difference, shedding light on the intricate molecular and contextual factors specific to esophageal adenocarcinoma.
Response: In accordance with the reviewer's advice, we have integrated the following passages into the discussion, addressing the role of the O-glycoproteome and exploring potential interactions between glycans and immune cells, including consideration of negative immune regulatory networks: “To decipher the composition of the O-glycome, the COSMC gene was selectively deactivated in numerous human cancer cell lines through gene editing techniques, inducing the expression of the Tn antigen. Subsequently, protein and glycopeptide isolation was achieved using the Tn antigen-affine plant lectin Vicia villosa agglutinin (VVL), followed by identification through mass spectrometric proteome analysis [35]. Notably, the discernment emerged that there exists no discernible overlap in the iden-tified O-glycoproteins, even when comparing cell lines within the same tumor entity. This revelation underscores the considerable heterogeneity inherent in the O-glycoproteome, introducing complexity into cell biological processes that defy straightforward predictions. Importantly, there remains a notable absence of an O-glycoproteome analysis for EAC to date.” and “Another pivotal consideration is the intricate interplay between different O-glycans within both the tumor and immune system cells. This dynamic interaction is critical in the context of glycoprotein modifications on the cell surface, influencing recognition by glycan-binding receptors such as C-type lectins, selectins, siglecs, and galectins. As glycans have evolved into essential components within homeostatic circuits, function-ing as precise modulators of immunological responses, they emerge as potential mo-lecular targets for intentionally modulating immune tolerance and activation across a spectrum of pathological scenarios. The recent review article emphasizes the signifi-cant role of Tn and STn antigen expression in gastrointestinal tumors in contributing to immunosuppression [49]. The focal point remains on innate immune system cells, specifically those expressing glycan-binding receptors like CD301 (MGL) [50], Siglec-4, -5, -14, and -15 [51], and potentially other receptors selectively interacting with tumor O-glycans.”
These revisions will enhance the manuscript by clarifying the clinical relevance of Tn and STn antigens in EAC and providing a nuanced understanding of why their prognostic implications might differ in this particular cancer type compared to others.
Round 2
Reviewer 1 Report
Comments and Suggestions for Authors
Well done.
Reviewer 2 Report
Comments and Suggestions for Authors I think the manuscript has been improved to warrant publication in Cancers. Comments on the Quality of English Language The quality of English can be further improved.